# Shear Strength of Industrial Wastes and Their Mixtures and Stability of Embankments Made of These Materials

**Andrzej Gruchot * and Tymoteusz Zydroń**

Department of Hydraulic Engineering and Geotechnics, University of Agriculture in Kraków,
al. Mickiewicza 24/28, 30-059 Kraków, Poland; tymoteusz.zydron@urk.edu.pl
\* Correspondence: rmgrucho@cyf-kr.edu.pl; Tel.: +48-12-6624136

**Abstract:** The paper presents the results of research on the influence of compaction on the shear strength of fly ash, unburnt and burnt coal wastes, and a composite (a mixture of unburnt coal waste and 30% of fly ash). The tests were carried out in a triaxial compression apparatus on samples with a diameter and height of 10 and 20 cm, respectively. In order to verify usability of the tested waste materials for earthworks, stability calculations of the road embankment made of these materials were carried out. It was shown that the tested materials were characterized by high values of shear strength parameters, which significantly depended on compaction. The most favorable values of the angle of internal friction and cohesion were obtained for the burnt coal waste, slightly lower for the composite, and the lowest for the fly ash. Stability calculations for the road embankment model showed that the slope inclination and the load on the embankment have a significant influence on the factor of safety. It was also shown that a decrease in cohesion causes significant decrease in the factor of safety. The tests and the stability calculations showed that the tested waste materials are useful for earth construction purposes.

**Keywords:** fly ash; coal waste; factor of safety

## 1. Introduction

Wastes are substances or objects that are disposed of, intended to be disposed of, or required to be disposed of. One of the types of waste is industrial waste, which is produced as a result of industrial activity of man and in many cases might be a valuable man-made aggregate. In the case of industrial waste, the main method of disposal is landfilling. This method may pose a significant threat to the environment, especially during improper landfilling or if the landfill location is wrong. The best solution is when the largest amount of waste is being managed. Currently, it is earth construction, including road construction, which allows for disposal of the largest amounts of industrial waste [1–5].

Waste generated during extraction and combustion of coal is the type of waste which is most commonly used in earth construction. Coal waste is a mixture of rock pieces generated during coal mining. Depending on the place where the waste is created, there are wastes from direct coal production and wastes that are produced during coal beneficiation (coarse-grained and fine-grained post-flotation waste). The petrographic composition of waste is mainly dominated by clay stones, clay shales, mudstones, coal shales, sandstones and, less often, conglomerates and hard coal fractions. Grains and particles of waste have a sharp-edged shape, which translates into higher shear strength compared to a material with a regular, round shape [5,6]. Coal waste is mainly used directly as a raw material for earthwork construction or land leveling. However, it should be properly compacted. Otherwise, infiltrating water from precipitation can contribute to the dissolution of chloride and increase in salinity, which can have a negative influence on the quality of both surface and groundwater.

Coal waste contains significant amounts of carbon and pyrite, which oxidize on exposure to air (atmospheric oxygen). Then an endoenergetic reaction occurs, the consequence of which is self-heating and, as a consequence, self-ignition of coal waste [7]. The temperature contributes to changes in the mineral and petrographic composition, and consequently it modifies characteristics and parameters of the burnt waste, which becomes a cheap and valuable material in earth construction. The properties of this transformed material meet the requirements for aggregates used in road construction [5,8].

Fly ash is another industrial waste generated in the process of burning ground hard coal or lignite in conventional or fluidized furnaces of power plants and combined heat and power plants. The ash structure is distinguished by its aggregate structure, with a complex surface, diverse porosity, and shear strength [9]. Fly ash is a secondary raw material used as a component of ceramic mass in the production of construction ceramics, as a component in the production of concrete and cement, for land leveling and land reclamation, and filling mining pits, as well as to mix with coal waste to prevent self-ignition. It is also used in mechanical stabilization of coarse soils to improve their granulation as well as a hydraulic binder that improves the mechanical parameters of road subsurfaces, especially in the case of forest roads in areas with high acidification [10]. Fly ash from brown coal combustion has binding properties and can be used as a concrete additive, but also as an independent hydraulic binder or as an additive in the production of these binders. Ash can also be used to stabilize some swelling soils, by reducing soil plasticity and swelling [11–13].

Mixing industrial waste in different proportions is a common way to increase the amount of wastes used in earthworks. By using waste mixtures, you can easily improve their chemical and physical properties, as well as increase their scope of use. By mixing coal waste and fly ash in fixed proportions, the parameters of the final product, such as impermeability, load-bearing capacity, or shear strength, improve compared to the initial ones [5,13–16]. Composites from ashes and coal wastes can be used in earthworks to build banks or rail and road embankments. Another application is surface hardening, land reclamation and land leveling, or using in underground mining techniques and ceramics industry [5,17,18].

Sherwood [2], Skarżyńska [6,19], and Gruchot [5] show that the geotechnical properties of coal waste depend on the grain size composition, retention time, and petrographic composition. Weathering, compaction, and moisture content have a great influence on the value of the angle of internal friction and cohesion. On the other hand, in the case of fly ash, its geotechnical properties will be related to the properties of the energy source, technology, and method of combustion as well as storage time [5,20,21]. Shear strength parameters of industrial wastes are usually favorable. Nevertheless, there are cases where waste embankments failed [22], which indicates that the use of industrial waste for construction purposes requires proper recognition of their geotechnical properties, mainly their shear strength parameters. As Skarżyńska [19] and Cadierno et al. [23] indicate, the basic problem in the case of coal waste testing is its grain size composition, and thus a significant content of large rock pieces. The presence of coarse grains makes it necessary to use medium- or large-size apparatus.

In this regard, it has to be noted that application of anthropogenic soils to construct earth-made structures requires two things: achieving suitable geotechnical parameters and fulfilling all environmental requirements. Unburnt colliery spoils are prone to a self-ignition process [19,24–26], which poses a threat to the structures made of them and contributes to emission of harmful chemical compounds to the atmosphere. Studies of leachates from colliery dumps indicate that these materials can contain sulphur and chloride compounds and some heavy metals [19,27–29], which can pollute the environment. An analysis of the leachate from structures built from colliery spoils in the area of southern Poland showed [4] that the amount of leachate can be reduced through proper execution of earthworks, mainly by high compaction of the structure. Consequently, high compaction of the earth-made structure reduces the amount of leachates. Fresh mine stones are usually neutral or alkaline, but during the weathering process an oxidation of pyrite occurs and they become acid. Fly ashes from ground coal, which are usually alkaline, show slightly distinct properties. Tests on the content of basic and trace elements, pH, and conductance of energy wastes confirm their high usefulness in

earthworks [30]. Zawisza's [31] test results indicate that mixtures of colliery spoils and fly ashes can provide a composite whose pH value can be neutral. Studies of fuel ashes from a few heat and power plants [5,32,33] indicate that their chemical composition depends on the type of coal, the amount and type of unburnt parts, and technical parameters of the power plant equipment. Studies of the leachate from industrial wastes indicate that the amount of leached substances did not exceed the limit values [30,33–35], so these wastes should not pose a threat to the quality of surface and underground waters. It can, therefore, be assumed that they won't endanger the natural environment around the structure. However, because of the high variability of their chemical parameters, an evaluation of their usefulness based on geotechnical tests and evaluation of their influence on the natural environment based on physicochemical tests must be carried out every single time.

The purpose of this study was to determine the influence of compaction on the shear strength parameters of selected industrial wastes. The following materials were used in the study: Fly ash from the electrostatic precipitator in Cracow Combined Heat and Power Plant, unburnt coal waste from current coal production and burnt coal waste from a heap in Mysłowice-Wesoła coal mine with a grain size smaller than 20 mm, and a composite (a mixture) of unburnt coal waste from Mysłowice-Wesoła and fly ash from Cracow Combined Heat and Power Plant (30% of fly ash in relation to the dry matter). Slope stability calculations were also carried out for a road embankment model made of the tested materials. The influence of cohesion changes on the factor of safety was taken into account during these calculations.

## 2. Test and Calculation Methods

The range of tests included determination of the grain size distribution, specific density, and compaction parameters using Polish standard (PN) or European standard (EN) (PN-EN, PN-EN ISO) methods [36–39]. Shear strength parameters, i.e., the angle of internal friction and cohesion, were determined using the UU (unconsolidated undrained) method in a triaxial compression apparatus. During the test the sample was compressed axially until the limit state was reached while maintaining a constant lateral pressure and constant deformation speed. Test specimens with a diameter of 10 cm and a height of 20 cm (Figure 1a) were formed in a three-part cylinder in a Proctor apparatus at optimum moisture content and unit compaction energy of 0.59 J·cm$^{-3}$ until the compaction indexes (values of relative compaction) $I_S = 0.90$ and 1.00 were reached. The value of Is parameter (also called relative compaction) expresses the ratio of the dry density of soil determined in the field tests (or for moulded samples in laboratory tests) to the value of the maximum dry density determined in a laboratory test (Proctor's method). After compaction, the samples were consolidated for 30 min and then sheared at the velocity of 0.33 mm min$^{-1}$ and at the following normal stresses: 50, 100, 200, 300, and 400 kPa. The shear criterion was the maximum value of the stress deviator in the range of up to 20% of the vertical deformation of the sample.

Knowing the values of shear strength parameters of materials used in earthworks is important, especially regarding construction and stability analysis of embankments. Stability analysis of road embankments is one of the main tasks that has to be done in order to verify their safe operation. The value of the factor of safety may vary depending on the method and above all on the adopted geotechnical parameters of soils in the embankment. Eurocode 7 [40] requires checking the GEO (geotechnical) or STR (structural) limit state, and thus showing that the design effects of the actions (turning moments) are smaller than the corresponding design soil resistance (stabilizing moments) determined based on design geotechnical parameters. With this approach, the minimum factor of safety should not be less than 1. The GEO limit state is associated with damage in the soil massif, e.g., in the form of a landslide. On the other hand, the STR limit state concerns cases of damage or large displacements in the soil massif with construction elements.

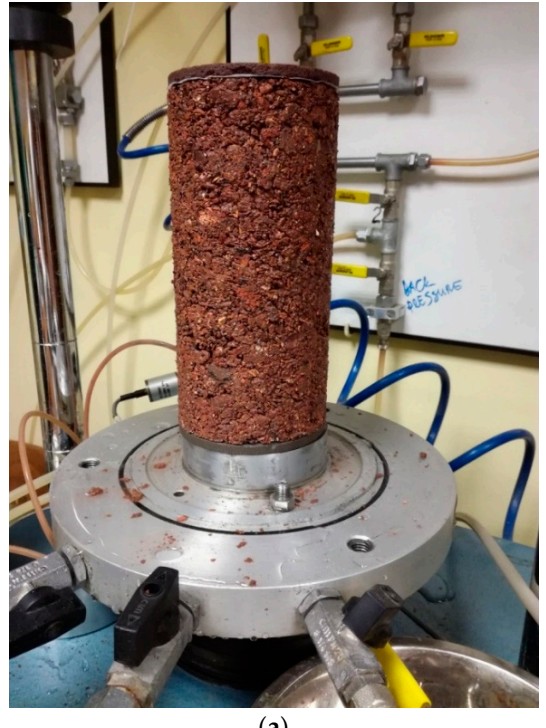
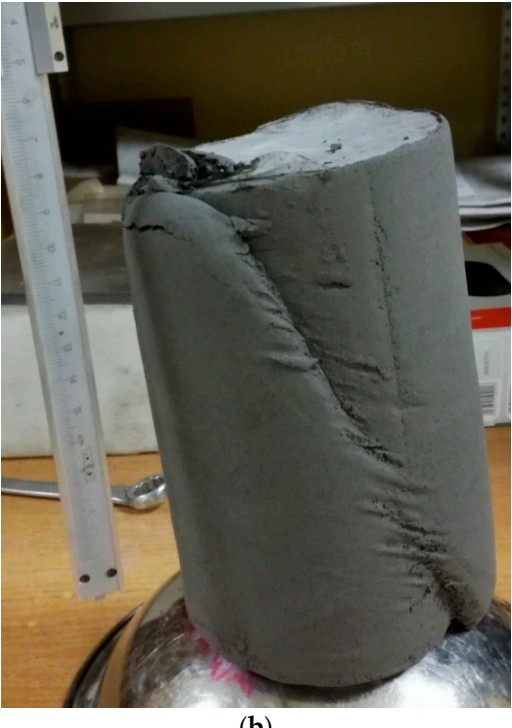

(**a**)  (**b**)

**Figure 1.** A formed sample of burnt coal waste (**a**) and a sample of fly ash after the test with a visible shear surface (**b**) (photo by A. Gruchot).

The limit equilibrium methods that are used in stability calculations allow determining the connection between the slope inclination, the load capacity, and the stress state in the entire massif. It is assumed that a limit stress state occurs along certain slip surfaces. Based on the premise that the deformation or destruction mechanisms occur along the slip surface, the force system associated with this mechanism was analyzed. Stability analysis led to the determination of the minimum value of the factor of safety, while taking into account the design values of geotechnical parameters as well as actions and resistances obtained as a result of using partial coefficients.

Stability calculations were carried out for a road embankment model with a height and crown width of 10 m at four slope inclinations: 1:0.5, 1:1, 1:1.5, and 1:2 (Figure 2). The calculations were made with the load on the embankment crown (evenly distributed vertical pressure of 10, 25, and 50 kPa) and without any load on the crown. It was assumed that the embankment was built from the fly ash, unburnt coal waste, and the composite (the compaction index was $I_S = 1.00$). Shear strength parameters and bulk density were adopted on the basis of obtained test results. In order to show the influence of cohesion on the factor of safety, it was assumed that the value of cohesion will correspond to the one obtained from the tests, but it will also be reduced to 50% and 10% of its initial value. Whereas the embankment subsoil was built of medium sand (MSa), the degree of compaction was $I_D = 0.70$ (the degree of compaction $I_D$, also called relative density, expresses the ratio of soil dry density determined in field tests to the value of maximum dry density of soil determined in laboratory test by vibration method). The calculations were carried out according to Eurocode 7 assumptions, adopting the DA3 calculation approach [28]. The calculations focused on the GEO limit state, assuming possible damage to the embankment and deformation of the subsoil in its vicinity. The calculations were carried out using the Janbu method in the Geo5 software.

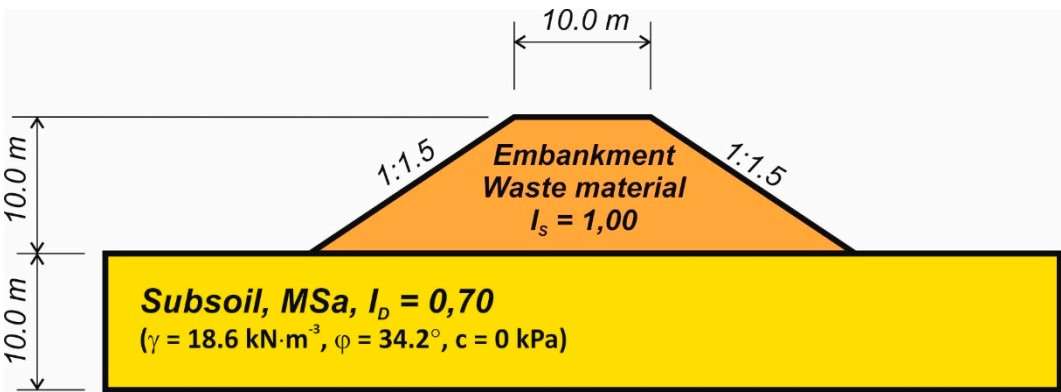

**Figure 2.** Scheme of the calculation model of the embankment with slope inclination of 1:1.5.

## 3. Test Results

### 3.1. Physical Parameters

The results of tests on physical properties of industrial waste are presented in Table 1, and Figure 3 shows the grain size curves of the tested materials. The grain size composition of the tested waste deserves a detailed discussion. Changes in the grain size composition due to mechanical disintegration resulting from the material compaction or weathering caused changes in the values of shear strength parameters [2,6]. The analysis of the grain size composition of the tested waste before and after the test (the compaction index $I_S$ = 1.00) showed significant differences in the content of each fraction.

The grain size composition of the finest material, i.e., fly ash, was dominated by the silt fraction, which was over 78%, the sand fraction was 17%, and the clay fraction nearly 5% (Table 1). According to the geotechnical nomenclature, grain size composition of the fly ash corresponded to well-graded medium silt.

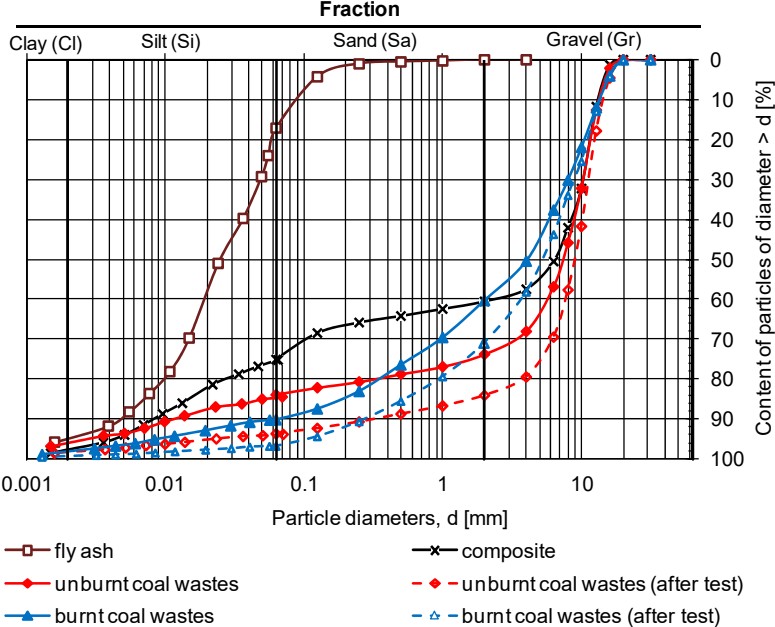

**Figure 3.** Grain size curves of the materials.

On the other hand, the grain size composition of coarse unburnt coal waste was dominated by the gravel fraction, which was 84%. The sand fraction (nearly 10%) and silt fraction (about 5%) were in much smaller quantities, and the clay fraction content was the smallest (1.5%). However, in their grain

size composition after the test, the content of gravel fraction decreased by 10%. In turn, the content of other fractions increased, by about 8% of the silt fraction, 1.5% of the clay fraction, and by nearly 1% of the sand fraction. According to the geotechnical nomenclature, these wastes, in terms of grain size composition, corresponded to well-graded medium gravels (before the test) and clayey medium gravels (after the test). Similarly, the composition of the burnt coal waste was dominated by the gravel fraction, which was 71%, while the sand fraction was 26%. On the other hand, the silt and clay fractions before the test were just over 3%. However, after the test, the content of gravel fraction decreased by 11%, and the sand, silt, and clay content increased by 4%, 5.5%, and over 1%, respectively. These wastes were classified as well-graded sandy medium gravels, before and after the test. It can, therefore, be stated that the tested coal wastes were susceptible to mechanical disintegration, mainly because of compaction, and, to a much lesser extent, because of shearing or grinding resulting from grain rotation during shearing of the sample.

The grain composition of the composite was dominated by the gravel fraction (61%). There was almost 15% of the sand fraction, 23% of silt, and slightly over 2% of clay. According to the geotechnical nomenclature, the composite was a well-graded silty medium gravel.

The specific density of the ash was the lowest and equaled 2.40 g·cm$^{-3}$, whereas in the case of the burnt coal waste it was the highest, 2.78 g·cm$^{-3}$. However, the specific density of the unburnt coal waste and the composite was similar and amounted to 2.25 and 2.33 g·cm$^{-3}$, respectively.

The maximum dry density of the fly ash was the lowest, 1.16 g·cm$^{-3}$ at the optimum moisture content of 34%, which was the highest among the tested materials. However, the maximum dry density of the coal waste and the composite ranged from 1.69 to 1.76 g·cm$^{-3}$ at the optimum moisture content from 6.7% to 17.6%. Similar values, as well as changes in compaction parameters along with the addition of fly ash were also noticed by Skarżyńska [6], Sherwood [2], Gruchot [5], and Blajer et al. [18]. It should be stated that the compactability of all wastes met the requirements for earth materials used for road construction in Poland [41].

**Table 1.** Geotechnical characteristics of tested waste materials.

| Parameter | Fly Ash | Coal Mining Wastes | | | | Composite |
| | | Unburnt | | Burnt | | |
| | | Before | After | Before | After | |
| | | Test | | | | |
| Fraction content [%]: | | | | | | |
| –gravel, Gr: 63 ÷ 2 mm | 0.0 | 84.1 | 73.8 | 71.0 | 60.4 | 60.6 |
| –sand, Sa: 2 ÷ 0.063 mm | 17.1 | 9.6 | 10.2 | 25.9 | 29.9 | 14.5 |
| –silt, Si: 0.063 ÷ 0.002 mm | 78.4 | 4.8 | 13.0 | 2.6 | 8.0 | 22.8 |
| –clay, Cl: <0.002 mm | 4.5 | 1.5 | 3.0 | 0.5 | 1.7 | 2.1 |
| Uniformity coefficient [-] | 7.5 | 33.5 | 741.7 | 84.3 | 25.0 | 1000.0 |
| Coefficient of curvature [-] | 1.5 | 13.2 | 114.7 | 2.3 | 2.5 | 0.2 |
| Name of soil acc. to [42] | MSi | MGr | clMGr | saMGr | | siMGr |
| Density of solid particles [g·cm$^{-3}$] | 2.40 | 2.25 | | 2.78 | | 2.33 |
| Optimum moisture content [%] | 34.00 | 6.65 | | 17.60 | | 13.40 |
| Maximum dry density of solid particles [g·cm$^{-3}$] | 1.160 | 1.760 | | 1.715 | | 1.690 |

*3.2. Shear Strength*

Analysis of the connection between the shear stress and vertical deformation showed that in most samples there was a brittle shear (Figure 4). However, it should be clearly indicated that in the case of samples where compaction index was $I_S = 0.90$, it was a shear that can be defined as ductile. In these tests, after reaching the maximum value, the shear stresses stabilized without a clear break in the stress–strain curve. Although at higher compaction, i.e., $I_S = 1.00$, in the case of burnt coal waste and

fly ash (Figure 1b), a brittle shear occurred. In this case, the shearing of the samples occurred when their vertical deformation was from 5% to 8%. A ductile shear was noticed only for burnt coal waste and the composite (Figure 4).

The values of the maximum shear stress depended on the type of soil and the main stress $\sigma_1$. When the main stresses ranged from 50 to 300 kPa, the shear stress values ranged from 315 to 990 kPa (compaction index $I_S = 0.90$) and from 320 to 1356 kPa ($I_S = 1.00$). For instance, when the main stress was 150 kPa and compaction index $I_S = 0.90$, there were similar values of shear stress for the fly ash and the composite and 2 times higher for burnt waste. However, when the compaction index was $I_S = 1.00$, the fly ash, unburnt waste, and the composite had similar values of shear stress, and burnt waste had values about 35% higher.

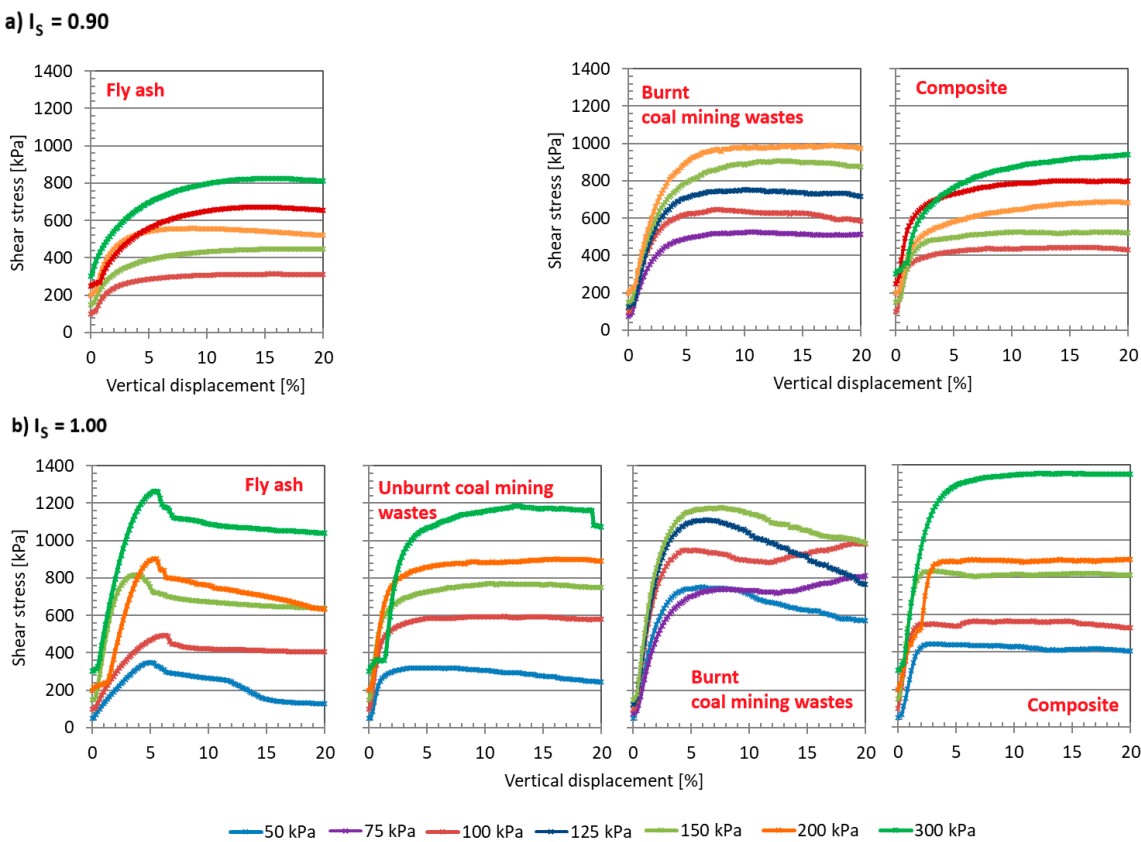

**Figure 4.** Changes in shear stress along with increasing axial deformation of samples.

Figure 5 shows the shear strength line as a tangent to Mohr's circles. The analysis of the drawings allowed us to state that the accuracy of the tests was satisfactory. This is particularly important considering that the majority of tested materials should be classified as coarse-grained soils.

The highest values of shear strength parameters were obtained for burnt coal waste, and the lowest for fly ash (in case of both compaction indexes). The tests showed that compaction has a significant influence on the tested parameters. In case of unburnt coal waste, the analysis of this influence was not carried out. This was due to the fact that the sample after forming (the compaction index $I_S = 0.90$) crumbled while putting it in the chamber of the triaxial compression apparatus.

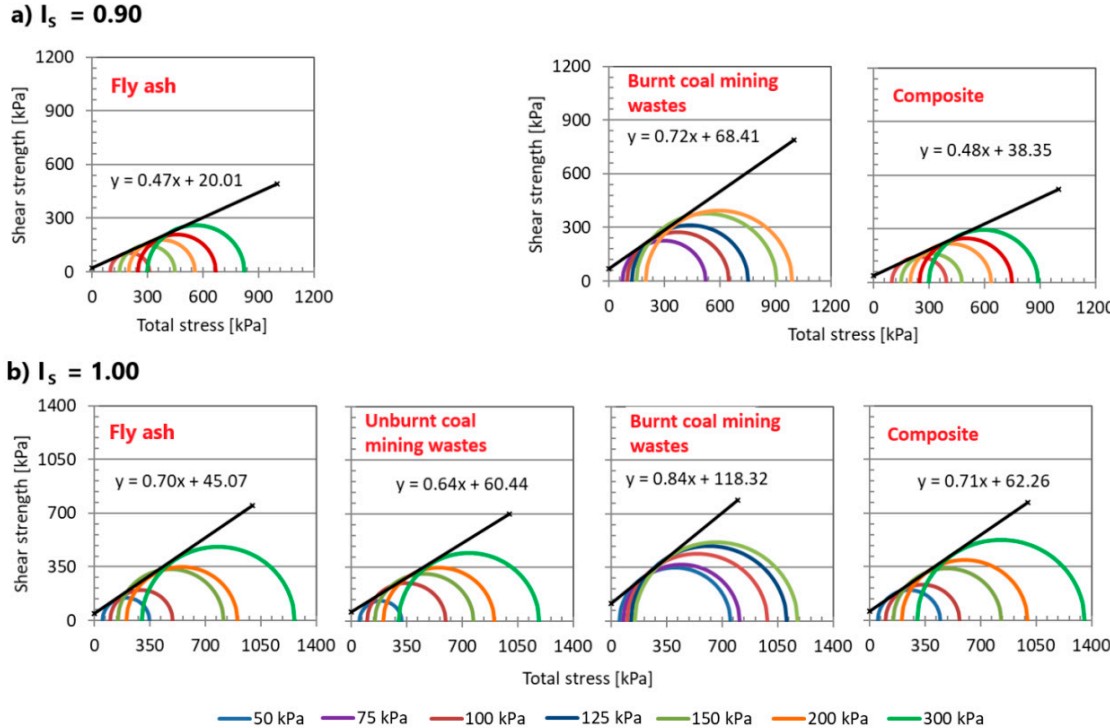

**Figure 5.** Shear strength line of the tested materials as a tangent to Mohr's circles.

The increase in compaction from $I_S = 0.90$ to 1.00 increased the angle of internal friction of the fly ash by nearly 10° (39% relative), of burnt coal waste by just over 4° (12% relative), and of the composite by nearly 9° (34% relative) (Figure 6). However, along with increase in compaction, the cohesion increased by 25 kPa (126% relative) for fly ash, by 50 kPa (73% relative) for burnt coal waste, and by 8.5 kPa (16% relative) for the composite.

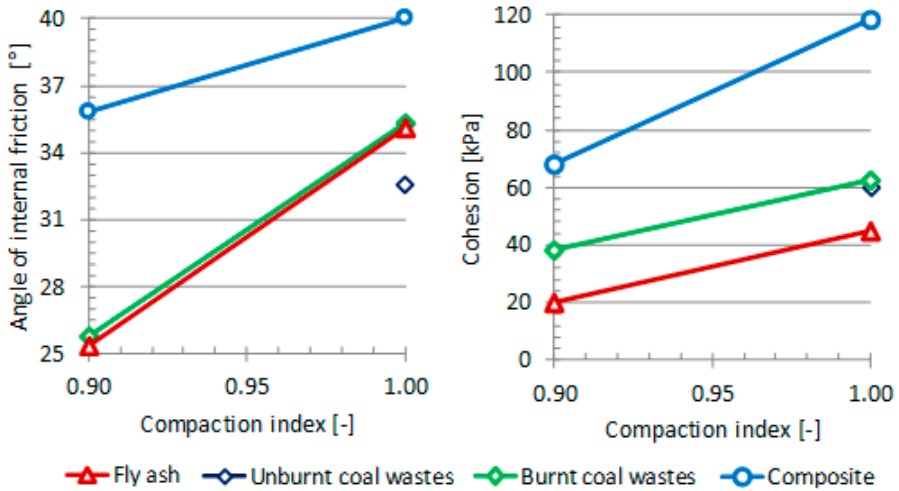

**Figure 6.** Influence of compaction on the angle of internal friction and cohesion.

By analyzing the influence of the fly ash addition to unburnt coal waste (Figure 7) it was found that when compaction index was $I_S = 1.00$ the angle of internal friction of the composite was higher by 2° (6% relative) compared to the value for unburnt coal waste and lower by 0.5° (1.5% relative) compared to the value for the fly ash. On the other hand, the cohesion of the composite at this compaction increased by 2.1 kPa (3% relative) and by 17.4 kPa (28% relative), respectively, in relation to the values for coal waste and the fly ash. When the compaction index was $I_S = 0.90$ the value of the angle of

internal friction of the composite was greater by 0.5° (2% relative) and cohesion by 34 kPa (63% relative) compared to the values for fly ash.

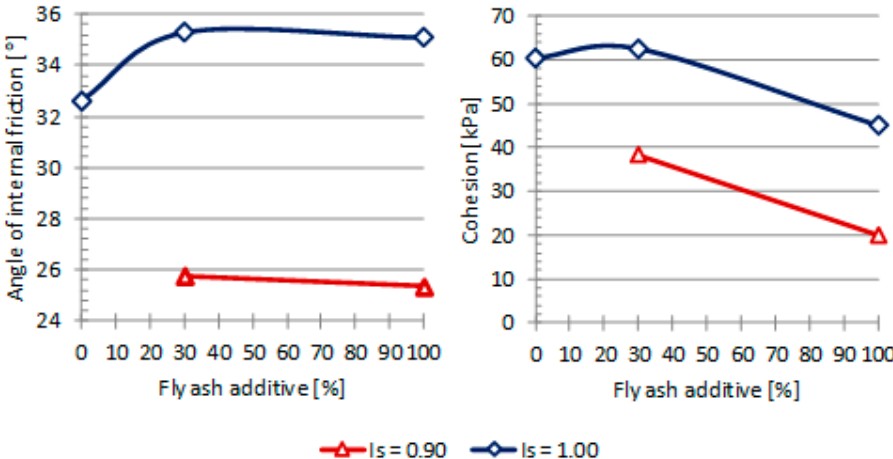

**Figure 7.** Changes in the angle of internal friction and cohesion depending on the fly ash addition.

Tests on coal wastes from the Upper Silesian Coal Basin (Poland) (among others: [4,5,19,43]) which were carried out using the direct shear method indicate that these wastes are characterized by high values of shear strength parameters. The highest values of the angle of internal friction (about 40°) were usually obtained for wastes from current coal production. This results from the sharp-edged shape of rock pieces and their surface roughness. In the case of weathered waste stored on landfills, the angle of internal friction was smaller by 5° compared to the value for waste from current production. In the case of treated wastes, the values of the angle of internal friction were different and ranged from 35° to 39°. Triaxial testing on coal wastes [32] also confirmed high values of shear strength parameters. The obtained results indicate that the values of these parameters depend significantly on the shear velocity and waste compaction. The authors showed that the increase in compaction from $I_S = 0.90$ to 1.00 increased the value of the angle of internal friction from 1° to 5°, which is similar to the relationship obtained in the presented tests on burnt coal waste. Gruchot et al. [44] also showed that the increase in shear velocity from 0.05 to 0.10 mm·min$^{-1}$ increased the angle of internal friction from 1° to 10° and the cohesion from about 45 to nearly 130 kPa.

Coal wastes from the Lublin Coal Basin (Poland) are characterized by slightly different geotechnical properties. Results of tests on these wastes with different storage time (from fresh waste to wastes stored in heaps for 5 and 7 years), using the direct shear method, presented by Flipowicz and Borys [45] confirmed the influence of the progressing weathering process on the values of the angle of internal friction and cohesion. As storage time increased, the content of coarse fractions decreased, which reduced the angle of internal friction. On the other hand, higher cohesion values, in relation to the tested waste and the waste from the Upper Silesian Coal Basin, were caused by a higher content of fine fractions, particularly silt and clay. The values of the angle of internal friction of these wastes ranged from 27° to 55°, and cohesion from 21 to 40 kPa.

In comparison, the results of tests on coal wastes from Great Britain [19,46] indicate that their angle of internal friction was in a fairly wide range (from 22.5° to 50°), and the cohesion was from 0 to 80 kPa.

The results of the presented fly ash tests showed that the values of the angle of internal friction ranged from 25° to 35°, which is a much wider range of values than those obtained by Zawisza and Zydroń [47] for fly ash from the 'Łęg' Heat and Power Plant in Cracow (35.0°–36.5°), however it falls within the range of values obtained for the type 'F' fly ash, studied by Kim et al. [48]. On the other hand, values of cohesion presented by Zawisza and Zydroń [47] were from 16 to 30 kPa, which can be considered as similar to those obtained in this work. Gimhan et al. [20] studied fly ash from a

coal power plant and they showed that at optimal moisture content the angle of internal friction and cohesion were 32.7° and 26.4 kPa. The authors of these studies indicate that fly ash is highly useful in civil engineering. Bera et al. [49] also showed high values of shear strength parameters of this type of waste (36° and 32 kPa).

Composites of unburnt coal waste with fly ash in the amount of 10%, 20%, and 30% by weight were the subject of Zawisza's research [50]. Studies have shown that the angle of internal friction was about 32° for the fly ash and it was slightly lower than the value for coal waste (just over 33°). For composites, the values of the angle of internal friction were higher and with the increase of ash content from 10% to 30% they decreased from 39° to 34°. Cohesion of fly ash was 33 kPa and was lower than for coal waste (47 kPa). In the case of composites, Zawisza and Blak [50] stated that as the ash content increased, cohesion decreased from 49 to 27 kPa. Blajer et al. [18] indicate that the values of shear strength parameters of coal waste and fluidized fly ash mixtures significantly increased with the addition of ash. However, they indicate that the values of the angle of internal friction were in a small range (from 35° to 41°), but with a significant increase in cohesion (from 37 to 55 kPa) when the ash addition increased from 0 to 40%, which was confirmed by the test results presented in this paper.

As shown above, the tested materials are characterized by high values of shear strength parameters, especially cohesion. This is consistent with previous results of test on industrial wastes (among others: [5,18,19]). However, it should be clearly emphasized that the tested materials, with the exception of fly ash, should be considered as coarse-grained, sharp-edged soils. In these types of materials, cohesion is not the result of intermolecular interaction, but interlocking of grains. Using such high cohesion values in design calculations may raise some concerns because some elements might be overestimated, for example, inclination of slopes at a given load on a road embankment. Therefore, in the further part of this work, stability calculations of a road embankment were carried out with reduced values of cohesion and unchanged values of the angle of internal friction.

## 4. Results of Stability Calculations

The factor of safety of the road embankment, calculated using the Janbu method, was in a fairly wide range, from 0.51 to 2.43. All the factors used in the calculations had an influence on this range, i.e., the inclination of the slope, load on the embankment, and the assumed cohesion value.

It should be clearly emphasized that the slope stability was also influenced by the angle of internal friction and bulk density of the material that was used to build the model. In these calculations, the values of the angle of internal friction were in a fairly narrow range, from about 33° to 35°. Therefore, it was assumed that this parameter will not be taken into account in further analysis. The next parameter was the bulk density, whose values for coal waste and the composite were similar. There was a significant difference only for the fly ash (reduction by nearly 3.5 kN·m$^{-3}$). In the further part of the analysis, it was decided not to take into account the influence of this parameter on the factor of safety.

The analysis of the test results showed that the slope of the road embankment would be unstable if the inclination was 1:0.5 and 1:1 with and without load on the embankment and assuming that the cohesion would be equal to 10% of the value from the tests. In these cases, the factor of safety ranged from 0.51 to 0.95 (Figure 8). In the other ones, the values of the factor of safety were much higher than 1. So, according to Eurocode 7, it can be stated that the slopes of all calculation models were stable.

It should be emphasized that as the cohesion increased, the range of the slope factor of safety increased for the adopted inclinations and loads on the embankment. When the cohesion was the same as the value obtained from tests in a triaxial compression apparatus the factor of safety ranged from 1.38 to 2.43, at 50% of its value from 0.96 to 2.01, and at 10% of its value from 0.51 to 1.51.

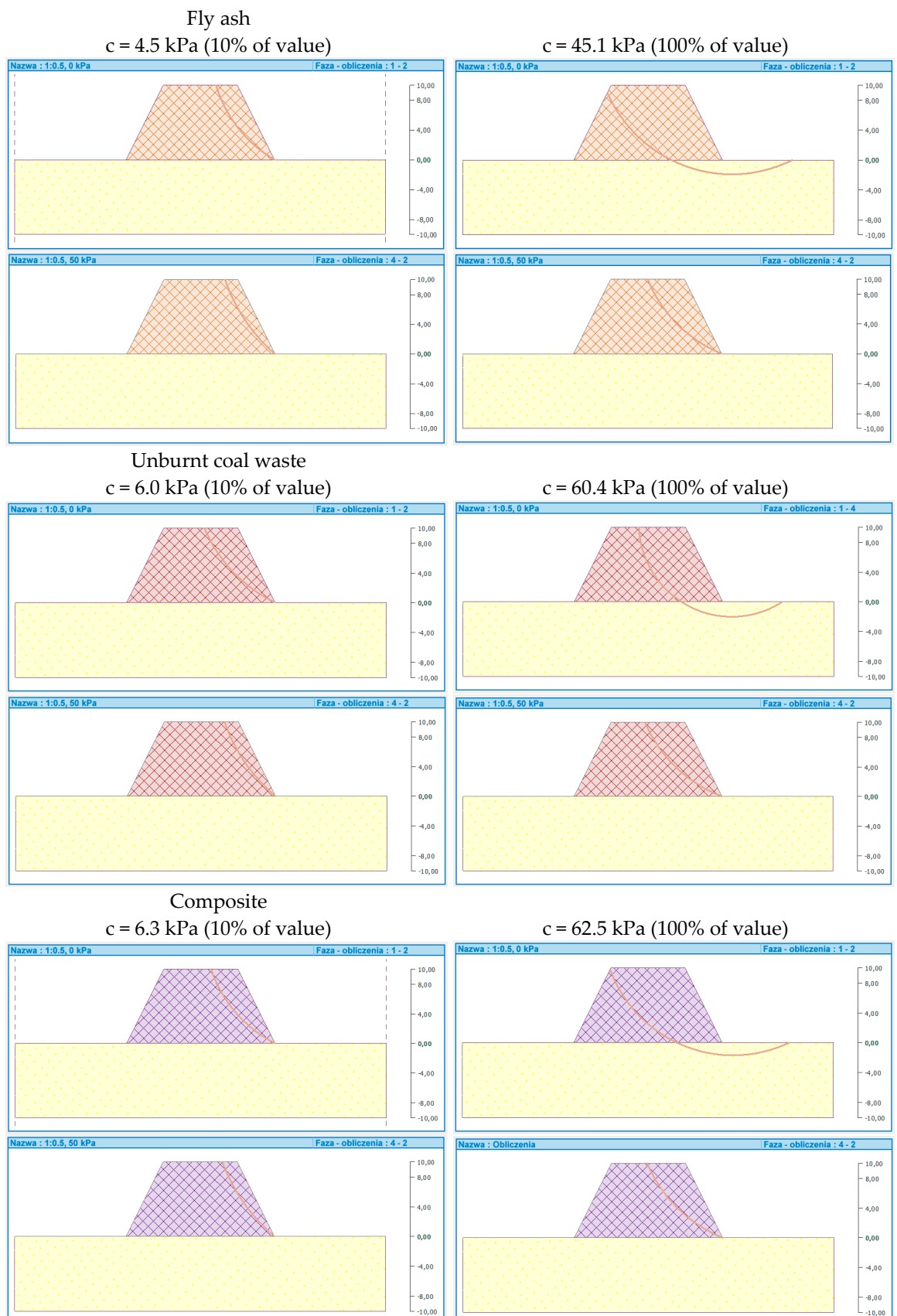

**Figure 8.** Slip surface at different load and cohesion values when the inclination was 1:0.5.

Increasing the load on the road embankment from 0 to 50 kPa resulted in a decrease in the factor of safety from about 14 to 39% (Figure 9) (adequately when cohesion was 10% of its test value and the maximum value obtained from the tests). For example, when cohesion was 10% of its test value there was a decrease in the factor of safety from about 0.65 to 0.52 (when the inclination was 1:0.5) and from 1.51 to 1.31 (when the inclination was 1:2). However, when the full consistency value was used, there was a decrease in the factor of safety from 1.92 to 1.38 at 1:0.5 slope and from 2.43 to 1.92 at a 1:2 slope. The ranges of changes in the factor of safety along with an increase in the load on the embankment corresponded to the full range of changes for all three materials at a given inclination of the slope. The analysis of Figure 9 shows that these changes are similar for all materials at the adopted cohesion value and inclination of the slope. However, the range of the factor of safety depends on the inclination of the slope and increases with its decrease.

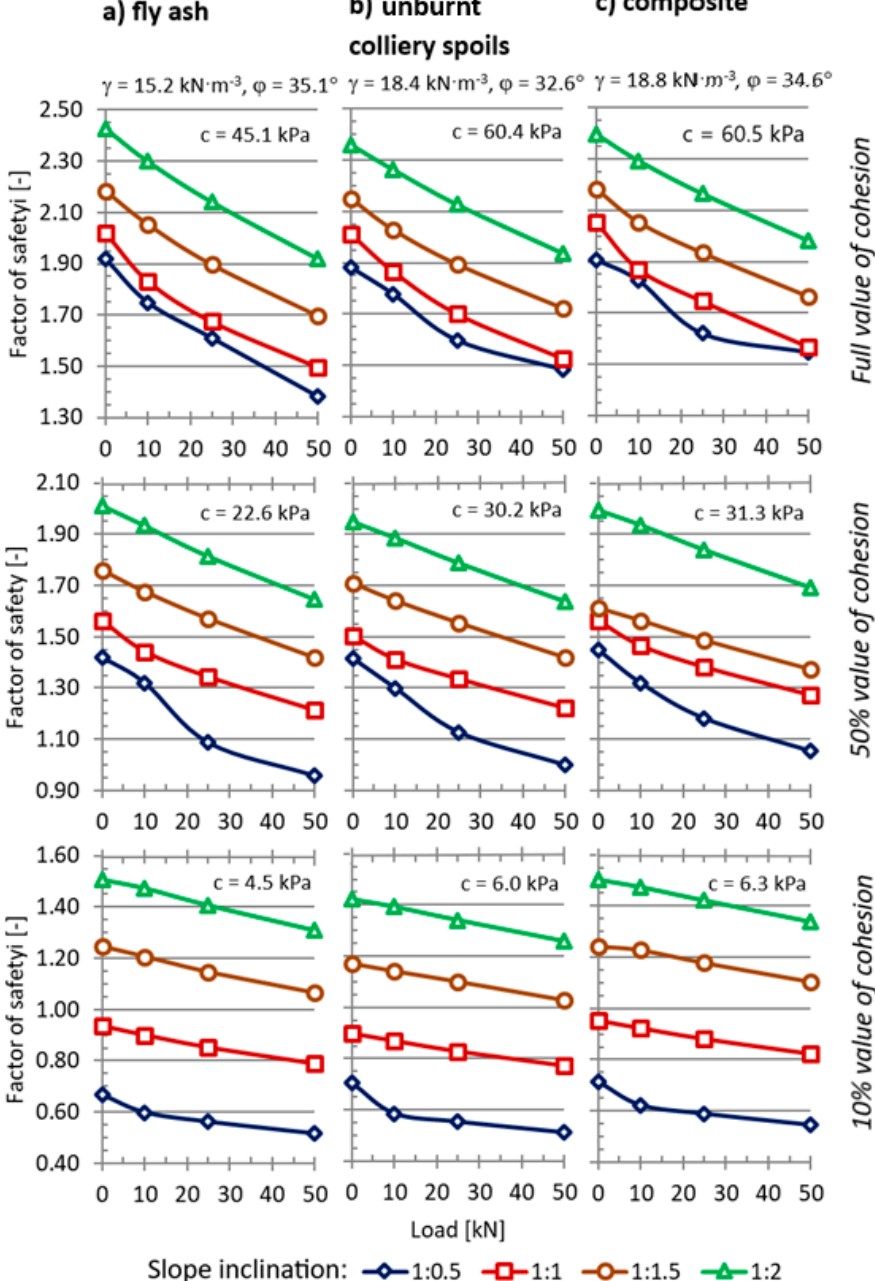

**Figure 9.** Changes in the factor of safety with increasing load on the embankment depending on the inclination.

As indicated previously, reducing the slope inclination from 1.05 to 1:2 resulted in a significant increase in the factor of safety (Figure 10). The range of these changes depended mainly on the cohesion value used in the calculations. At its full value, i.e., obtained from tests, the factor of safety increased on average by 0.51, at 50% of its value on average by 0.63, and at 10% of its value on average by 0.80 with decreasing inclination for all materials and all embankment loads. The values of increase of the factor of safety given above are an average value of the difference between its maximum and minimum value for a given material and load. Figure 10 shows an example of the relation between the factor of safety and inclination (while using the full consistency value, obtained from the tests).

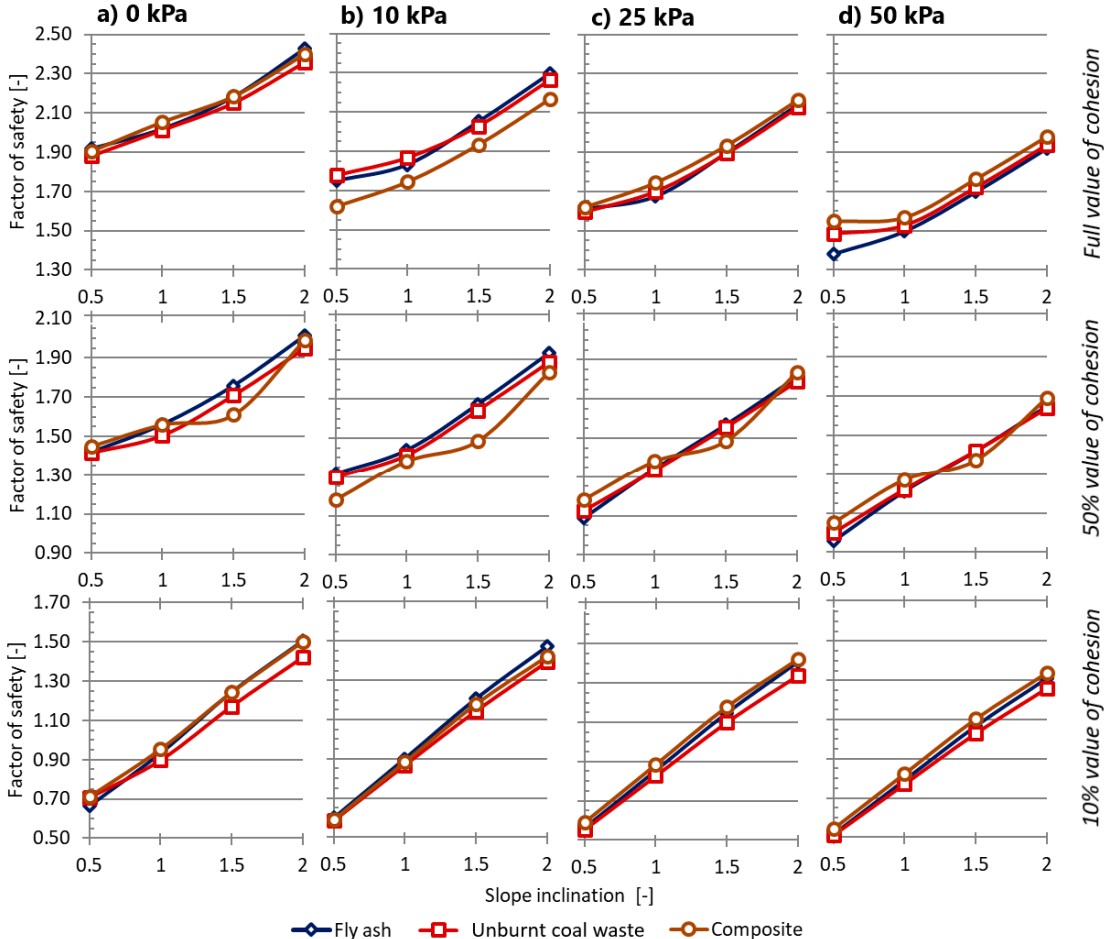

**Figure 10.** Changes in the factor of safety with decreasing inclination at a given load on the embankment.

It can, therefore, be concluded that cohesion plays an important role in the stability calculations (Figure 11). Slight differences between the values of the factor of safety of each material within calculations using the same cohesion value may result from small differences in the angle of internal friction and bulk density.

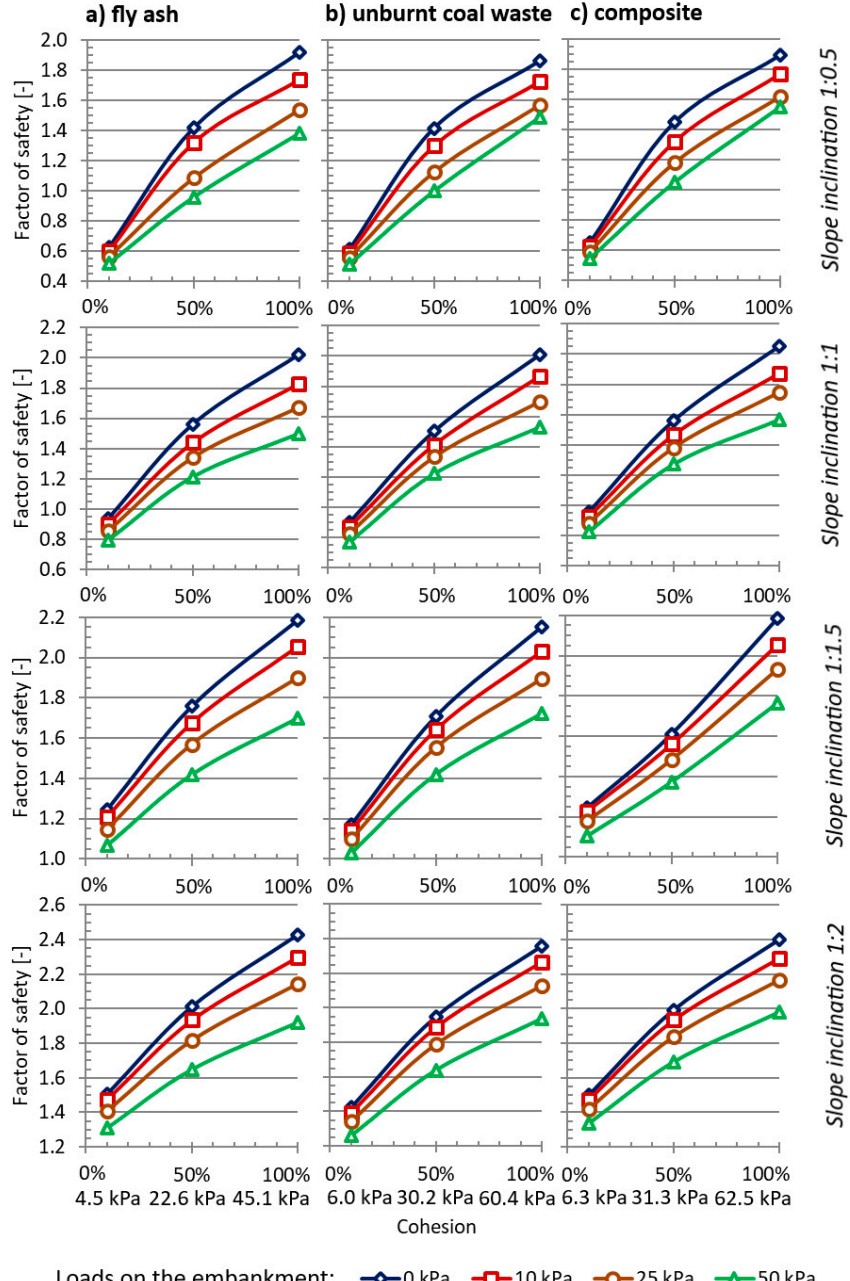

**Figure 11.** Changes in the factor of safety with increasing cohesion at different loads on the embankment.

Numerous publications (among others: [22,51–56]) indicate that the problem of slope stability in industrial waste landfills is an important engineering issue. The results of tests on shear strength parameters of materials stored there are often carried out either at low values of their moisture content or at optimum moisture content and indicate that these materials usually have high values of shear strength parameters, although the height of the slopes or their too-steep inclination are the reason why they are destroyed (a landslide occurs). These failure causes are an indication that detailed studies of geotechnical properties, reduction of the amount of wastes deposited on landfills, or changes in the slope inclinations should be done.

Another important element of waste management is using mixtures with other wastes or mineral soils. As the presented research indicate and, among others, research by Rajak et al. [57], the addition of fly ash increases the shear strength of natural soils. They stated that the factor of safety of a 37°

slope, obtained from calculations in the FLAC/Slope program, increased with the increase in the fly ash addition [57], indicating that the optimal addition of fly ash was 30% for a slope up to 12 m high.

A particular problem when using waste materials is the difficulty in interpreting their shear strength related to the occurrence of additional friction on contact between coarser grains manifested as the occurrence of so-called apparent cohesion [4,19], which is accepted or omitted in further analyses. The interpretation of shear strength is especially important at low normal stresses and, depending on the assumed shear criterion, can lead to significant discrepancies [58,59]. This is important because in case of stability calculations, cohesion plays an important role (Figure 11). It should also be mentioned that during operation there are different weather conditions so slopes of the earth embankments are alternately dry and wet. If these processes occur, it should be expected that coal wastes will disintegrate, which will lead to changes in their geotechnical properties. While comparing the results of tests on unburnt and burnt waste, it seems right to state that the strength of the material due to disintegration and thermal transformations has a positive effect on its properties. On the other hand, if only the second process occurs, it can lead to a decrease in cohesion. Therefore, it is justified to use reduced cohesion in stability calculations.

## 5. Summary

The subjects of this paper were shear strength parameters of industrial waste as well as calculations of slope stability of a road embankment model with different inclinations and variable load. The tests were carried out in a triaxial compression apparatus for wastes and their mixtures.

The tests showed that industrial wastes had high values of the angle of internal friction and cohesion, which confirms their usefulness in earthworks. It was shown that the values of these parameters depended on the compaction, grain size distribution, and, in particular, the content of coarse fractions. The most favorable values of these parameters were obtained for the burnt coal waste, and slightly less for the composite of unburnt coal waste and 30% of fly ash.

Stability calculations of the road embankment model showed that the inclination and the load have a significant influence on the factor of safety values. It was also shown that changes in the cohesion value, which in these tests were big, also significantly reduced the factor of safety. A recommendation that the cohesion value is reduced by at least 50% should be used in the design and stability calculations of earth embankments from waste materials, in particular coarse-grained ones. The calculations have also showed that there was only a slight difference in the factor of safety for embankments made of each waste material, which indicates that their usefulness for road construction is similar. On the other hand, before they are used, some additional tests should be carried out, for example, in case of fly ash, chemical properties, since they can contribute to the swelling processes.

The performed tests and calculations related to the use of industrial waste in earthworks show the validity of their engineering management. Therefore, the tests will be continued, and their scope will be extended to include the influence of water and other chosen non-linear shear criteria, whereas the stability calculations will also take into account different dimensions of the embankment.

**Author Contributions:** Conceptualization, A.G.; methodology, A.G.; software, A.G.; validation, A.G. and T.Z.; formal analysis, A.G. and T.Z.; investigation, A.G.; resources, A.G.; data curation, A.G. and T.Z.; writing—original draft preparation, A.G.; writing—review and editing, A.G. and T.Z.; visualization, A.G.; supervision, A.G.; project administration, A.G.; funding acquisition, A.G. and T.Z. All authors have read and agreed to the published version of the manuscript.

**Funding:** This research received no external funding.

**Conflicts of Interest:** The authors declare no conflict of interest.

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
