# Peer review of "Shear Strength of Industrial Wastes and Their Mixtures and Stability of Embankments Made of These Materials"

_applsci, doi:10.3390/app10010250_

Round 1

Reviewer 1 Report

Applied Science

Title: Shear strength of industrial wastes and their mixtures and stability of embankments made of these materials

Dear authors, the topic you are exploring is very interesting, the application of different waste materials: flying ash unburned and burned coal and a mixture of waste as material for road embankment. In order to use these materials, they should satisfy the strength and slope stability – You have made it clear in the manuscript. So the manuscript is well structured and clearly written.

Since there are no results in the paper on the environmental impact of these materials, I think you should give these results, the chemical composition of the materials used and what is the effect of rain on the leaching of toxic substances from these materials. I think that environmental impact is a very important element in applying this type of waste for these purposes.

Author Response

We thank you for the positive review.

Reviewer 2 Report

Dear Authors, suggestion is to add more journal references of peer reviewed ones where environmental considerations are mentioned as well as environmental safety (leaching etc.) should be at least marginally issued in the paper

Author Response

Point 1:

Suggestion is to add more journal references of peer reviewed ones where environmental considerations are mentioned as well as environmental safety (leaching etc.) should be at least marginally. 

 Response 1:

Thank you for the review.
A note regarding the description of the impact of industrial waste on the natural environment in the result of possible leachate has been taken into account.
This fragment was included in the “Introduction” of the manuscript and highlighted in red.

Point 2:

English language and style are fine/minor spell check required

Response 2:

The translation of the manuscript was prepared by a specialized translation agency.
The manuscript was also reviewed by a native-speaker.